# A Hybrid PAPR Reduction Scheme in OFDM-IM Using Phase Rotation Factors and Dither Signals on Partial Sub-Carriers

**DOI:** 10.3390/e24101335

**Published:** 2022-09-22

**Authors:** Si-Yu Zhang, Hui Zheng

**Affiliations:** 1Key Laboratory of Modern Measurement and Control Technology, Ministry of Education, Beijing Information Science and Technology University, Beijing 100101, China; 2School of Information and Communication Engineering, Beijing Information Science and Technology University, Beijing 100101, China

**Keywords:** index modulation (IM), dither signals, peak-to-average power ratio (PAPR), orthogonal frequency division multiplexing (OFDM), bit error rate (BER), energy efficiency

## Abstract

As a multi-carrier modulation technique, a high peak-to-average power ratio (PAPR) is a common issue suffered by orthogonal frequency division multiplexing with index modulation (OFDM-IM) due to its system structure. High PAPR may cause signal distortion, which affects correct symbol transmission. This paper tries to inject dither signals to the inactive (idle) sub-carriers, which is a unique transmission structure of OFDM-IM, to reduce PAPR. Unlike the previous works, which utilize all idle sub-carriers, the proposed PAPR reduction scheme utilizes selected partial sub-carriers. This method performs well in terms of bit error rate (BER) performance and energy efficiency, which are obvious drawbacks of the previous PAPR reduction works due to the introduction of dither signals. In addition, in this paper, phase rotation factors are combined with the dither signals to compensate for the PAPR reduction performance degradation due to the insufficient use of partial idle sub-carriers. Moreover, an energy detection scheme is designed and proposed in this paper in order to distinguish the index of phase rotation factor used for transmission. It is shown by extensive simulation results that the proposed hybrid PAPR reduction scheme is able to implement an impressive PAPR reduction performance among existing dither signa-based schemes as well as classical distortion-less PAPR reduction schemes. In addition, the proposed method obtains better error performance and energy efficiency than that of the previous works. At the error probability $10^{-4}$, the proposed method can achieve around 5 dB gain compared to the conventional dither signal-based schemes

## 1. Introduction

The novel and advanced communication technologies attract increasing attention and interests from universities and industry, in order to meet the explosive data traffic and massive digital service requirements. Orthogonal frequency division multiplexing (OFDM), as a multi-carrier transmission technique, has become one of the most popular techniques in 4G long-term evolution (LTE) communication systems [1,2]. OFDM has impressive advantages in terms of frequency selective fading resilience and low-cost hardware implementation. However, in 5G networks, due to the massive connected devices and explosive growth in mobile data services, the implementation of new selection techniques becomes necessary [3]. The novel techniques should have higher spectral and energy efficiency with ultra reliability. Therefore, to further improve the throughput and decrease the energy consumption, based on existing techniques, the novel concept of index modulation (IM) has been proposed.

OFDM-IM [4] is a promising technique extended from the concept of the index modulation technique in spatial domain [5]. Conveying information through the indices of idle sub-carriers is an unique property of OFDM-IM. In OFDM-IM, the information bits are transmitted through two statuses, the constellation symbols and the indices of sub-carriers. An overview of this technique can be found from works [6,7]. Compared with the conventional OFDM, OFDM-IM can achieve a trade-off between energy efficiency and system performance with an adjustment of the number of active sub-carriers in the system [8]. OFDM-IM has been applied in several scenarios, such as vehicle to vehicle (V2V) networks [9] and visible light communication systems (VLC) [10]. Therefore, OFDM-IM is a promising candidate for the next generation communication systems thanks to its fitness and flexibility for some business in beyond 5G (B5G) or 6G [11].

However, as a kind of multi-carrier modulation technique, OFDM-IM also has a significant weakness of high peak-to-average power ratio (PAPR), which is inherited from OFDM and general multi-carrier techniques. High PAPR may raise non-linear distortion and error performance degradation [12] when OFDM-IM signals pass through high power amplifiers (HPAs). Hence, if no PAPR reduction scheme is implemented, OFDM-IM signals can cause serious problems, including a severe power penalty at the transmitter, which is particularly not affordable in portable wireless devices. For conventional OFDM, there are several PAPR reduction schemes utilizing different domain resources, such as clipping, pre-coding [13], active constellation extension (ACE) [14], selective mapping (SLM) [15], partial transmit sequence (PTS) [16], tone reservation (TR), etc. These PAPR reduction schemes can be directly implemented in OFDM-IM systems and have been verified to achieve satisfying performance. However, the unique properties of OFDM-IM systems are not considered, which may limit the corresponding PAPR reduction performance of OFDM-IM.

Compared with the conventional OFDM, OFDM-IM owns a number of idle (inactive) sub-carriers, which may be used for PAPR reduction. In order to take advantage of the idle sub-carriers of OFDM-IM systems, the author in [17] proposed an efficient PAPR reduction scheme by injecting some amplitude constraint dither signals, which are generated from convex programming, to the idle sub-carriers. This paper can be regarded as the first work utilizing the structure of OFDM-IM. According to this work, several schemes and algorithms are investigated and proposed to further explore idle sub-carriers for improving PAPR reduction performance [18,19,20] by utilizing multi-level or time-domain dither signals. However, all of the above works do not solve the problem of bit error rate (BER) degradation due to the introduction of dither signals. Moreover, these extra signals actually affect the energy efficiency of OFDM-IM, which compromises an irreplaceable advantage of OFDM-IM compared with OFDM.

In this paper, to solve the problems of error performance and energy efficiency degradation in previous works of PAPR reduction based on dither signals, a hybrid PAPR reduction method that jointly utilizes dither signals and phase rotation factors is proposed. The proposed method injects dither signals to partial idle sub-carriers in order to make a trade-off between PAPR reduction performance and error performance. Simulation results and theoretical analyses show that utilizing partial sub-carriers is helpful for enhancing error performance and energy efficiency, but the PAPR reduction performance is compromised. Therefore, to compensate the PAPR reduction performance degradation, the proposed scheme introduces phase rotation factors before the injection of dither signals. These phase rotation factors are helpful to reduce PAPR. Such hybrid system can select one signal with the minimum PAPR for transmission from a number of candidates with different PAPR thanks to the introduction of phase rotation factors. Moreover, in order to detect the index of the phase rotation factor selected by the transmitted signal, in this paper, a detection method is also proposed based on energy detection. It is shown by extensive simulation results that the proposed PAPR reduction method can obtain better error performance and energy efficiency than that of the previous dither signal-based works. Furthermore, by utilizing the phase rotation factor, the proposed PAPR reduction scheme can also achieve a superior PAPR reduction performance than previous schemes.

Conclusively, the contributions of this study can be summarized and highlighted as follows:This paper proposes a novel PAPR reduction method based on dither signals. The proposed method injects optimized constant amplitude signals on partial inactive sub-carriers. Comparisons show that utilizing partial idle sub-carriers can achieve better error performance than that of the conventional dither signal-based schemes.In order to solve the PAPR reduction performance degradation caused by the insufficient use of idle sub-carriers, the proposed PAPR reduction scheme introduces phase rotation operations. Simulation results show that compared with the conventional works, the proposed hybrid scheme can achieve better performance in terms of both PAPR and BER.This paper proposes an energy-based detection method in order to recover the transmitted signal. The proposed detection method is able to correctly detect the index of the transmitted phased rotation factor. Simulation results show that the decoding performance, which utilizes the proposed detection method is very close to the case where the side information is correctly received. Therefore, without sending side information, the spectral efficiency of OFDM-IM is higher than that of the previous works.

The remaining part of this paper is organized as follows: Section 2 provides the preliminary knowledge, including OFDM-IM, PAPR, and the conventional PAPR reduction with dither signals as a background introduction that is required for the following part of this paper. The proposed hybrid PAPR reduction scheme and the corresponding analysis are given in Section 3. Simulation results are provided in Section 4, and the conclusion part is given in Section 5.

## 2. Preliminary Knowledge

### 2.1. OFDM with Index Modulation

For an OFDM-IM system, information bit sequences *X* with length *m*, are first split into *G* sub-blocks. Each sub-block $X_{g}$ contains *p* bits, thus m=pG. Each sub-block is mapped to an OFDM sub-block of length *n*, where N=nG is the number of available sub-carriers. OFDM-IM is not only performed by means of modulated symbols but also by the indices of sub-carriers in a modulated sub-block [4].

The illustration of an OFDM-IM system is given in Figure 1. For each sub-block, $p_{1}=\left\lfloor \log_{2}C_{n}^{k}\right\rfloor$ bits are utilized to choose *k* active sub-carriers from *n* available sub-carriers, where $C_{a}^{b}$ denotes the binomial coefficient. Here, the set of the indices of the *k* active sub-carriers in the *g*th sub-block is denoted as: $I_{g}=\left\{i_{g}\left(1\right),i_{g}\left(2\right),\cdots ,i_{g}\left(k\right)\right\}$, where $i_{g}\left(\gamma \right)\in \left\{1,2,\cdots ,n\right\}$ for g=1,2,⋯,G and γ=1,2,....,k. The rest of the $p_{2}=k\log_{2}M$$\left(p_{1}+p_{2}=p\right)$ bits are modulated to *M*-ary constellations. Therefore, the *g*th set of modulated symbols can be denoted as $S_{g}=\left[S_{g}\left(1\right),S_{g}\left(2\right),\cdots ,S_{g}\left(k\right)\right]$, where $S_{g}\left(\gamma \right)\in S$ and *S* is the constellations. By considering $I_{g}$ and $S_{g}$ for all *G* sub-blocks, an OFDM block X can be written as:(1)$$X={\left[X\left(1\right),X\left(2\right),\cdots ,X\left(N\right)\right]}^{T}$$
where N=Gn, X(α)∈{0,S}, for α=1,2,⋯,N. *T* denotes the transpose. The pattern for active sub-carrier selection can be completed by the look-up table method [4]. An example of such an active sub-carriers selection for an n=4, k=2 sub-block is given in Figure 2. It can be seen from Figure 2, that if the input information bit of the sub-block is [110001], when M=2, the first two bits [11] are index bits, and the set of the indices is I={2,4} according to the table. The remaining four bits [0001] are mapped according to the constellation and carried by the second and fourth sub-carriers of the block. Therefore, OFDM-IM transmits four bits of information by utilizing only two sub-carriers, which improves the energy efficiency and inter-carrier interference (ICI) resilience.

Therefore, by considering $p_{1}$ and $p_{2}$, we can conclude from the total bits that an OFDM-IM block is sent as:(2)$$G\left(p_{1}+p_{2}\right)=G\left(\left\lfloor \log_{2}C_{n}^{k}\right\rfloor +klog_{2}M\right)$$

According to the transmission rule, the spectral efficiency of OFDM-IM is [21]:(3)$$R_{ofdm-im}=\frac{\left(p_{1}+p_{2}\right)}{n}=G\left(\frac{\left\lfloor \log_{2}C_{n}^{k}\right\rfloor +k\log_{2}M}{N}\right)$$

After, an inverse discrete Fourier transform (IDFT) block is applied to the baseband-equivalent OFDM-IM symbols, which can be written as:(4)$$x=F^{H}X$$
where *H* represents the Hermitian transpose, F denotes the unitary DFT matrix. Then, the cyclic prefix (CP) is appended, and parallel to serial (P/S) and digital to analog (D/A) conversions are followed.

At the receiving side, after the removal of CP, a DFT operation needs to be implemented as the conventional OFDM. However, unlike the conventional OFDM, the OFDM-IM receiver needs to detect both the indices of active sub-carriers and the corresponding modulated symbols. For each sub-block, by considering a joint detection for the indices of active sub-carriers and the transmitted symbols carried on, the maximum likelihood (ML) detector for OFDM-IM is given by:(5)$${\left\{{\hat{S}}_{g}\left(\gamma \right)\right\}}_{\gamma =1}^{n}=\underset{{\left\{S_{g}\left(\gamma \right)\right\}}_{\gamma =1}^{n}}{\mathrm{argmin}}\sum_{\gamma =1}^{n}{\left|Y_{g}\left(\gamma \right)-H_{g}\left(\gamma \right)S_{g}\left(\gamma \right)\right|}^{2}$$
where $H_{g}\left(\gamma \right)$ denotes the channel frequency response (CFR) on the γth sub-carrier of the *g*th sub-block. $Y_{g}\left(\gamma \right)$ is the γth received signal after CP removal and DFT. More details for ML decoding in OFDM-IM can be found in [22]. The searching space of the ML detection per bit is given by the order of O(nk+Mk) [23]. Hence, the optimal ML detector has high complexity. To achieve near-optimal performance, two low-complex detectors have been proposed, including the low-complex ML and the a posteriori probability detection methods [24]. In this paper, an energy-based detection method is proposed in order to find the correct phase rotation factor used for PAPR reduction, and the ML detection is then utilized to find the transmitted symbols.

### 2.2. PAPR

In this sub-section, the concept of PAPR is introduced. The PAPR of a vector x is defined as [12]:(6)$$\Gamma_{x}=\mathrm{PAPR}\left\{x\right\}≜\frac{\underset{1\leq \alpha \leq N}{\max}{\left|x\left(\alpha \right)\right|}^{2}}{P_{x}}$$

In the above equation, the average power of x is written as $P_{x}$. The corresponding calculation for $P_{x}$ is written as: $P_{x}=\frac{1}{N}E\left[x^{H}x\right]=\frac{1}{N}{E[||x||}^{2}]$, where E[] denotes the expected value. In practical scenarios, the performance of PAPR reduction is usually measured by the complementary cumulative distributive function (CCDF) of the corresponding PAPR [25]. The definition of CCDF is: CCDF is the probability that the PAPR of a block x exceeds a given threshold $PAPR_{0}$ (usually given in decibels), given as:(7)$$\begin{array}{cc} F_{\Gamma_{x}}^{c}\left(PAPR_{0}\right) & =1-F_{\Gamma_{s}}\left(PAPR_{0}\right)\\ \; & =1-Pr[\mathrm{PAPR}\left\{x\right\}\leq PAPR_{0}]\\ \; & =Pr[\mathrm{PAPR}\left\{x\right\}>PAPR_{0}] \end{array}$$
where $F_{\Gamma_{x}}\left(PAPR_{0}\right)=Pr\left[\Gamma_{x}\leq PAPR_{0}\right]$ denotes the cumulative distributive function of $\Gamma_{x}$.

### 2.3. The Conventional PAPR Reduction with Dither Signals

In this sub-section, the classical PAPR reduction scheme with constant amplitude dither signals [17] is briefly introduced.

Figure 3 is a block diagram that illustrates a PAPR reduction scheme utilizing constant amplitude dither signals. It is assumed that the set of indices of the inactive sub-carriers of X is denoted as $I^{c}$, where *c* means the complement. We note that the cardinality of is $I^{c}$, N−K, where K=kG. In Figure 3, for a dither signal-based PAPR reduction scheme in frequency domain, N−K dither signals with amplitude constraint *R* are injected to the N−K inactive sub-carriers. Since dither signals are random, the injected signals can be seen as an extra noise to the original signals. Therefore, small constant *R* is regulated as the maximum amplitude of the dither signals in order to control the impact on demodulation errors. It is shown in (Equation 6) that minimizing the PAPR of x is equivalent to minimizing the numerator of (Equation 6), which is equivalent to minimizing the infinity norm of the vector x. Therefore, according to (Equation 6), the optimal dither signals can be found through the convex programming. The PAPR optimization problem for such a method can be written as:(8)$$\begin{array}{c} \min_{\zeta }\left|\right|F_{I^{c}}^{H}\zeta +x{\left|\right|}_{\infty }^{2}\\ {s.t.||\zeta ||}_{\infty }\leq R \end{array}$$
where the input time-domain signal vector is denoted as $x=F^{H}X$. $F_{I^{c}}^{H}\in C^{N\times (N-K)}$ represents the matrix $F^{H}$ by deleting the columns whose indices belong to *I*. The dither signals in the frequency domain is written as $\zeta \in C^{(N-K)\times 1}$. The ζ can be obtained through the CVX package [26], which is a specific algorithm package used for convex programs.

The illustration of the super constellation after injection dither signals is given in Figure 1 of [17]. It is illustrated that despite the fixed signals for the active sub-carriers, the dither signals on idle sub-carriers with the signal amplitude constraint *R* are around the origin, which can be regarded as a kind of adding noise that compromises both error performance and efficiency.

Then, the frequency domain symbol vector $X_{d}$, which is injected by the optimized dither signals, are implemented by IDFT operations. In practical scenarios, the operations of IDFT are usually replaced by inverse Fast Fourier transform (IFFT).

At the receiving side, both maximum likelihood (ML) [4] and the log-likelihood ratio (LLR) detector can be applied for the active sub-carrier [27] detection. The detection procedure of the dither signal-based PAPR reduction scheme is similar to the conventional OFDM-IM, where the extra dither signals can be treated as a kind of adding noise. The signal mode (active or inactive) of each sub-carrier can be determined by looking at the logarithm of the ratio between the a posteriori probabilities of the sub-carriers being modulated by the information symbols and zero values, respectively, which is formulated as:(9)$$\lambda \left(\alpha \right)=\ln\frac{\sum_{j=1}^{M}Pr\left(X\left(\alpha \right)=S\left(j\right)|Y\left(\alpha \right)\right)}{Pr(X(\alpha )=0|Y(\alpha ))}$$
where the size of constellation *S* is denoted as *M*. Y(α) is the αth (1≤α≤N) received signal after inverse operations (CP removal and DFT). It is obvious that if λ(α) is larger than 0, the αth sub-carrier is more likely to be modulated by information symbols instead of being inactive. We note that since the inactive sub-carriers are occupied by small amplitude signals instead of zeros, the error performance of the dither signal-based PAPR reduction schemes are usually degraded.

## 3. The Hybrid PAPR Reduction Scheme with Phase Rotation Factors and Dither Signals on Partial Subcarriers

According to existing works, it is known that applying dither signals in OFDM-IM is an effective way to reduce its high PAPR. However, injecting PAPR reduction dither signals can be regarded as another kind of adding noise, which will inevitably affect error performance. Furthermore, adding extra signals may compromise the high energy efficiency advantage enjoyed by OFDM-IM. In this section, we first give an interference analysis of the PAPR reduction schemes based on dither signals. Then, we give a PAPR reduction scheme utilizing partial sub-carriers based on channel state information (CSI). This ideal method can be regarded as the best performance of the bit error rate that our proposed hybrid method can achieve. Then, the proposed hybrid PAPR reduction scheme with phase rotation factors and dither signals on partial sub-carriers is proposed. Last, we give the energy efficiency analysis of the proposed scheme compared with the existing works.

### 3.1. Interference Analysis

Consider a case that a symbol vector $X_{g}$ in sub-block *g* is erroneously determined to be ${\hat{X}}_{g}$. We assume that the optimized dither signals are $\zeta_{g}^{\prime }=\left[\zeta_{1,g},\zeta_{2,g},\cdots ,\zeta_{3,g}\right]$. Consider the fact that $Y_{g}=\mathrm{diag}\left\{X_{g}+\zeta_{g}\right\}H_{g}+W_{g}$, where H is the frequency channel response, $H={\left[H_{1},H_{2},\cdots ,H_{g}\right]}^{T}$, $H_{g}={\left[H_{g}\left(1\right),H_{g}\left(2\right),\cdots ,H_{g}\left(n\right)\right]}^{T}$, and W is the added White Gaussian noise (AWGN). The conditional pair-wise error probability (PEP) can be written as:(10)$$\begin{array}{cc} \; & P(X_{g}\to \hat{X_{g}}|H_{g},Y_{g},w/\mathrm{dithersignal})\\ \; & =P(||Y_{g}-\mathrm{diag}\left\{{\hat{X}}_{g}\right\}H_{g}|{|}^{2}\leq ||Y_{g}-\mathrm{diag}\left\{X_{g}\right\}H_{g}|{|}^{2})\\ \; & =P(||\mathrm{diag}\left\{X_{g}-{\hat{X}}_{g}\right\}H_{g}+\mathrm{diag}\left\{\zeta_{g}^{\prime }\right\}H_{g}+W_{g}|{|}^{2})\\ \; & \leq ||\mathrm{diag}\left\{\zeta_{g}^{\prime }\right\}H_{g}+W_{g}{\left|\right|}^{2} \end{array}$$
where the second equality is from the fact $Y_{g}=\mathrm{diag}\left\{X_{g}+\zeta_{g}^{\prime }\right\}+H_{g}+W_{g}$.

The error occurrence of $X_{g}\to {\hat{X}}_{g}$ can be categorized into two cases. The first case is that the system successfully detects the active sub-carriers, while the constellation symbols are erroneously detected. The second case is that $X_{g}$ and $\hat{X_{g}}$ have different detected active sub-carriers. It is obvious that when the amplitude of dither signals *R* is large, the idle sub-carriers are more likely to be detected as active ones. Therefore, we discuss two cases here according to the amplitude of *R*.

*Case 1, R is small*: If *R* is small, it is more likely for the system to detect the correct active sub-carrier set. In such a scenario, ${\hat{X}}_{g}\left(i\right)\neq 0$ if $X_{g}\left(i\right)\neq 0$ and ${\hat{X}}_{g}\left(i\right)=0$ if $X_{g}\left(i\right)=0$, for i=1,2,⋯,n. Moreover, if $X_{g}\left(i\right)\neq 0$, then $\zeta_{i,g}=0$, according to the rules of dither signal injection. Therefore, (Equation 10) is calculated as:(11)$$\begin{array}{cc} P( & ||\mathrm{diag}\left\{X_{g}-{\hat{X}}_{g}\right\}H_{g}+\mathrm{diag}\left\{\zeta_{g}^{\prime }\right\}H_{g}+W_{g}{\left|\right|}^{2}\\ \; & \leq ||\mathrm{diag}\left\{\zeta_{g}^{\prime }\right\}H_{g}+W_{g}{\left|\right|}^{2})\\ \; & =P(||\mathrm{diag}\left\{X_{g}-{\hat{X}}_{g}\right\}H_{g}+W_{g}|{|}^{2}\leq ||W_{g}|{|}^{2})\\ \; & =P(X_{g}\to {\hat{X}}_{g}|H_{g},Y_{g},w/o\;\mathrm{dithersignals}) \end{array}$$
which is equivalent to the conditional PEP without considering dither signals. Therefore, the conditional PEP is the same in both cases with and without the PAPR processing when *R* is small.

*Case 2, R is large*: However, when the amplitude of dither signal *R* is large, the interference of $\mathrm{diag}\left\{\zeta_{g}^{\prime }\right\}H_{g}$ cannot be ignored. When the signal-to-noise ratio (SNR) is high, the main interference becomes $\mathrm{diag}\left\{\zeta_{g}^{\prime }\right\}H_{g}$ instead of $W_{g}$. Therefore, the idle sub-carriers are more possible to be detected as an active one. Hence, the conditional PEP will be compromised with the increase of *R*, and an error floor can be found in [17] in high SNR regions due to the interference brought by dither signals.

Considering the fact that the dither signals can affect the error performance and energy efficiency of the system, a very straightforward idea for keeping such properties is to utilize partial instead of idle sub-carriers. Moreover, it can be expected that when utilizing the sub-carriers, which have low frequency channel responses, the error performance degradation can be further limited. Furthermore, by reducing the number of sub-carriers used for PAPR reduction, the error performance can be improved.

### 3.2. The PAPR Reduction Scheme Utilizing Partial Idle Sub-Carriers Based on CSI

The previous works that utilize dither signals explore all idle sub-carriers. Such schemes are able to keep solid PAPR reduction performances. However, from the perspective of error performance, these methods may not be the best. Therefore, we propose and analyze the PAPR reduction method, which utilizes partial idle sub-carriers according to the CSI. This scheme picks the idle sub-carriers whose own low channel gains add dither signals for PAPR reduction.

Figure 4 shows the transmitter structure of the proposed PAPR reduction scheme utilizing partial idle sub-carriers based on CSI (denoted as “S.1”). It is assumed that the frequency domain channel response $H={\left[H\left(1\right),H\left(2\right),\cdots ,H\left(N\right)\right]}^{T}$, and the percentage of partial idle sub-carriers that require dither signal injection is β. The indices for dither signals depend on the channel gain. Hence, in the S.1 scheme, the channel response H needs to be sorted first, which can be written as:(12)$$\tilde{H}={\left[\tilde{H}\left(1\right),\cdots ,\tilde{H}\left(\alpha \right),\cdots ,\tilde{H}\left(N\right)\right]}^{T}$$
where 1≤α≤N, and $|\tilde{H}\left(1\right)|<\ldots |\tilde{H}\left(\alpha \right)|<\ldots |\tilde{H}\left(N\right)|$. The system selects the sub-carriers with the least β(N−K), where K=Gk, channel gains for dither signal injection. If the indix set of positions which own the least β(N−K) channel gains is denoted as $I_{h}^{c}$, the PAPR optimization problem for this method can be written as:(13)$$\begin{array}{c} \min_{\zeta }\left|\right|F_{I_{h}^{c}}^{H}\zeta +x{\left|\right|}_{\infty }^{2}\\ {s.t.||\zeta ||}_{\infty }\leq R \end{array}$$

Symbol sequence X is generated from information bits after index modulation. The dither signal $d_{h}$ is obtained through convex programming. The positions of $d_{h}$ are according to the channel response H. Then, X and $d_{h}$ are added to the frequency domain. Finally, the transmitted time domain signal vector $x_{h}$ is obtained by an IDFT operation.

Figure 5 shows the BER performance among the proposed S.1 scheme with a different number of β (β∈{0.25,0.5}), the scheme utilizing dither signals in [17] (denoted as “Dither”), and the original OFDM-IM signals (the ideal case). It is illustrated from Figure 5 that the S.1 scheme utilizing partial idle sub-carriers according to CSI can achieve a very close performance to that of the original OFDM-IM signals. Furthermore, due to the injected noise, it is obvious that the more idle sub-carriers are utilized, the worse BER performance is achieved. Therefore, using partial idle sub-carriers for dither signals is an effective way for PAPR reduction without compromising the error performance of the system.

However, utilizing partial idle sub-carriers may compromise the PAPR reduction performance due to the insufficient use of idle sub-carriers. The PAPR reduction performance comparison among the proposed S.1 scheme and Dither [17] scheme is shown in Figure 6, using different number of β (β∈{0.25,0.5}) with N=128 sub-carriers. This result highlights that although using partial sub-carriers for dither signals can significantly improve the error performance, the PAPR reduction capability degradation compared with the conventional Dither scheme cannot be overlooked. Furthermore, obtaining an instantaneous channel response is not always available in practical cases thus, the S.1 scheme in this section can only be regarded as an ideal case. This ideal case can be regarded as the best error performance that our proposed hybrid method can achieve. In the next subsection, a hybrid PAPR reduction scheme that combines phase rotation factors is proposed. The proposed hybrid scheme can significantly improve the system’s PAPR reduction capability as well as keeping a solid error performance.

### 3.3. The Proposed Hybrid PAPR Reduction Scheme

Adding dither signals on corresponding partial idle sub-carriers is helpful to prevent error performance degradation. However, choosing partial indices for a dither signal injection could affect PAPR reduction performance. Therefore, in order to compensate for the PAPR reduction performance degradation due to the insufficient use of partial sub-carriers, this section introduces a hybrid method that implements phase rotation factors for modulated symbols to generate different candidates with different PAPR. The system can select one with the lowest PAPR to transmit.

Figure 7 shows the transmitting structure of the proposed hybrid PAPR reduction scheme. Symbol sequence X is generated from information bits after index modulation. The proposed scheme first defines an ensemble of phase rotation factors, as:(14)$$b_{u}={\left[b_{u}\left(1\right),b_{u}\left(2\right),\cdots ,b_{u}\left(N\right)\right]}^{T}$$
where u=1,2,⋯,U, that are known to both transmitter and receiver. The entries of $b_{u}$ are formed as follows:(15)$$b_{v}\left(\alpha \right)=e^{j\phi_{u}\left(\alpha \right)}$$
where $\phi_{u}\left(\alpha \right)\in \left[0,2\pi \right)$, α=1,2,⋯,N. In general, binary elements are randomly selected for building $b_{v}\left(\alpha \right)$, that is, $b_{v}\left(\alpha \right)\in \left\{+1,-1\right\}$. In the proposed hybrid method, the component-wise multiplication of the input symbol vector X with *U* different phase rotation factors are firstly implemented.

Then, *U* dither signals $d_{u}$ whose idle sub-carriers for injection are different from each other are injected into $Xb_{u}$. These *U* dither signals can be obtained from convex programming. Then, after signal injection, *U* parallel IDFT (or IFFT) operations are implemented to obtain *U* time domain candidates $x^{u}$. After obtaining *U* candidate sequences, the proposed PAPR reduction scheme chooses one candidate with the lowest PAPR for transmission. We note the corresponding signal as $x^{\tilde{u}}$ and $x^{\tilde{u}}$ can be selected based on the following equation:(16)$$x^{\tilde{u}}=\underset{u=1,2,\cdots ,U}{\underset{︸}{\mathrm{argmin}}}PAPR\left\{x^{u}\right\}$$

In order to improve the PAPR reduction performance and facilitate the detection procedure, the proposed scheme utilizes the “interleaved” method for idle sub-carriers’ partitioning. For simplicity, in this part, we consider an available idle indices set $I^{c}$ for all *N* sub-carriers, which is written as:(17)$$I^{c}=\left\{i_{1},i_{2},\cdots ,i_{N-K}\right\}$$
where K=Gk, and $i_{\gamma }\in \left\{1,2,\cdots ,N\right\}$, 1≤γ≤N−K. The idle index set needs to be divided into *V* disjoint subsets, such that:(18)$$I^{c}=\sum_{v=1}^{V}I_{v}^{c}$$
where:$$\begin{array}{cc} \; & I_{1}^{c}=\left\{\underset{(N-K)/V}{\underset{︸}{i_{1},i_{V+1}\;\ldots ,\;i_{N-K-V+1}}}\right\},\\ \; & I_{2}^{c}=\left\{\underset{(N-K)/V}{\underset{︸}{i_{2},i_{V+2}\;\ldots ,\;i_{N-K-V+2}}}\right\},\\ \; & \;\;\vdots \\ \; & I_{V}^{c}=\left\{\underset{(N-K)/V}{\underset{︸}{i_{V},i_{2V}\;\ldots ,\;i_{N-K}}}\right\} \end{array}$$

The idle sub-carriers set $\chi_{u}$ for *U* candidates can be jointly determined through the combination of these *V* sub-blocks and the idle sub-carrier active ratio β. To facilitate the detection procedure, in the proposed method, $\chi_{u}$ and $b_{u}$ have a one-to-one correspondence. Once *V*, β, and *U* are determined, the idle sub-carriers set $\chi_{u}$ can be determined by utilizing a look-up table. A mapping example between $b_{u}$ and $\chi_{u}$ with V=4, U=4, and β=0.5 is given in Table 1.

Here, we take an example where V=3. U=2, β=0.25, N=4 to illustrate how candidates with different PAPR are constructed. It is assumed that X=[X(0),0,0,0]. If $b_{1}=\left[1,1,1,1\right]$ and $b_{2}=\left[-1,-1,-1,-1\right]$, $\chi_{1}=\left\{I_{1}^{c},I_{3}^{c}\right\}=\left\{2,4\right\}$, $\chi_{2}=\left\{I_{2}^{c},I_{3}^{c}\right\}=\left\{3,4\right\}$, the construction of candidates $X^{1}$ and $X^{2}$ after corresponding dither signals injection can be illustrated in Table 2.

Therefore, the PAPR optimization problem for this hybrid method can be written as:(19)$$\begin{array}{c} \min_{\zeta }\left|\right|F_{\chi_{u}}^{H}\zeta +x{\left|\right|}_{\infty }^{2}\\ {s.t.||\zeta ||}_{\infty }\leq R \end{array}$$

At the receiving side, the selected index $\tilde{u}$ needs to be detected in order to recover the original transmitted signals. On the one hand, side information that represents which phase rotation factor b is used can be transmitted as extra bits. However, this method will transmit at least $\log_{2}U$ extra bits, which compromises the spectral efficiency. Moreover, the incorrect detection of side information bits may cause catastrophic error performance.

Here, we consider a low-complex energy-based detection scheme. First, the indices of idle sub-carriers are required to be detected as the conventional OFDM-IM scheme. Then, the transmitted $b^{\tilde{u}}$ are detected based on the energy distribution of the received sequence. Here, we denote the received sequence and the estimated idle indicex set as Y and ${\hat{I}}^{c}$, respectively. The detection process is summarized below.

Step 1: Obtain all possible idle sub-carrier sets ${\hat{\chi }}_{u}$ by utilizing ${\hat{I}}^{c}$ and look-up tables.

Step 2: Obtain the received signal blocks $Y_{{\hat{\chi }}_{u}}$ according to the received sequence Y and ${\hat{\chi }}_{u}$.

Step 3: Compute the signal energy $|Y_{{\hat{\chi }}_{u}}{|}^{2}$, for u=1,2,…,U and arrange them in descending order,
(20)$$|Y_{{\hat{\chi }}_{1}}{|}^{2}\geq |Y_{{\hat{\chi }}_{2}}{|}^{2}\geq \ldots \geq {\left|Y_{{\hat{\chi }}_{U}}\right|}^{2}$$
where *u* is the index of the *u*th largest energy signal block.

Step 4: Find the set corresponding to ${\hat{\chi }}_{1}$ as the idle sub-carriers set and determine the corresponding phase rotation factor b utilizing the look-up table.

The complexity of the energy-based detection method can be separated into two parts. First, the sorting complexity of Step 3 is O(U) if *U* candidates and appropriate sorting algorithms are considered. Then, the received signal corresponding to ${\hat{\chi }}_{1}$ is detected. If the LLR detection method is utilized, the corresponding complexity can be denoted as O(nkMk2) [28].

Figure 8 illustrates the index error probability (IEP) of the proposed energy-based detection method with various *U*, *V*, and β. It can be seen from the result that *U* and *V* are dominate factors that affect the IEP. It is reasonable that increasing *U* will make it more difficult for the detection method to identify the correct $\tilde{u}$. However, it will be shown in the next section that with the same β, increasing *U* will benefit the PAPR reduction performance.

### 3.4. Energy Efficiency of the Proposed Schemes

In this section, we briefly analyze the energy efficiency of the proposed scheme. The average transmit power per sub-block is denoted as $k+\beta \left(n-k\right)R^{2}$. The power loss η of the conventional dither signals scheme compared with the proposed hybrid scheme [17] is denoted as:(21)$$\begin{array}{cc} \eta & =10\log_{10}\frac{k+\beta \left(n-k\right)R^{2}}{k+\left(n-k\right)R^{2}}\\ \; & \approx \frac{10\left(1-\beta \right)\left(n-k\right)R^{2}}{[k+\beta \left(n-k\right)R^{2}]\ln10}\mathrm{dB} \end{array}$$

For a practical system with β=0.5, n=4, k=2, the power loss is around 4.34 $\left(1-\frac{2}{\left(2+R^{2}\right)}\right)$ dB, which is larger than 0. Therefore, it is obvious that the proposed scheme owns higher energy efficiency than that of the existing PAPR reduction schemes with dither signals.

## 4. Simulation Results

In this section, simulation results are presented in terms of PAPR reduction and BER performance. The simulation of the proposed hybrid PAPR reduction scheme using phase rotation factors and dither signals on partial sub-carriers is implemented. Scenarios with different *U* (U∈{2,4}), β (β∈{0.25,0.5}), and *V* (V∈{2,4}) are simulated and the corresponding results are compared with the conventional PAPR reduction scheme with dither signals [17] (denoted as “Dither”). As reference, the PAPR performances of PTS, SLM, the S.1 scheme, and the original signal without any PAPR reduction scheme are also included. For the OFDM-IM system, we use N=128, n=4, and k=2. QPSK symbols are applied. For fairness, we use R=0.5 for all selected schemes.

The PAPR reduction performance comparison among the proposed scheme, PTS scheme, SLM scheme, and Dither scheme in [17] is shown in Figure 9. This figure also compares the proposed scheme with different *U* (U∈{2,4}), β (β∈{0.25,0.5}), and *V* (V∈{2,4}). This result highlights that all PAPR reduction schemes have significant PAPR reduction capabilities. It is shown that the S.1 scheme described in the last section achieves the worst PAPR reduction performance due to the unexplored idle sub-carriers. However, by introducing phase rotation factors, significant PAPR reduction performance enhancement can be obtained. The proposed hybrid scheme can achieve flexible performance achieved by the conventional PTS and SLM schemes designed for OFDM, the proposed hybrid scheme can achieve an impressive performance in most scenarios. Moreover, it is shown that when U=4,V=4,β=0.5, the proposed hybrid scheme can achieve even better performance than that of the Dither scheme, which utilizes all idle sub-carriers. Furthermore, it is obvious that the more idle sub-carriers are used (e.g., β=0.5, β=0.25), the better the PAPR reduction performance of the system can be achieved.

Figure 10 shows the BER performance among the proposed scheme with different parameters, the Dither scheme in [17], and the original OFDM-IM signals. It is illustrated from Figure 10 that the proposed method achieves better BER performance than of the conventional dither signal-based scheme. When U=2, V=2, and β=0.5, the proposed scheme can achieve 5 dB gain compared with the conventional method at error probability $10^{-4}$. However, it is also shown in Figure 10 that although the proposed hybrid scheme owns better PAPR reduction performance, it still has BER degradation compared with the S.1 scheme, which achieves almost the same performance as that of the original OFDM-IM signals (the ideal case). Nevertheless, the proposed hybrid scheme can still obtain better BER performance results and higher energy efficiency compared with the Dither scheme. Additionally, it is shown that *U* and *V* can affect the error performance because *U* and *V* affect the IEP of the energy-based detection utilized by the proposed PAPR reduction scheme.

## 5. Conclusions

In this paper, a novel hybrid PAPR reduction scheme without sending side information for OFDM-IM is proposed by jointly utilizing phase rotation factors and dither signals on partial idle sub-carriers. The proposed scheme takes advantage of partial idle sub-carriers for PAPR reduction, which achieves a superior BER performance and energy efficiency compared with the conventional dither signal-based PAPR reduction schemes. In addition, this paper utilizes phase rotation factors to further improve the PAPR reduction performance. Moreover, since the index of the transmitted phase rotation factor needs to be detected, an energy-based detection method is proposed. By proposing the energy-based detection method, the proposed scheme does not require side information bits transmission, which further improves its spectral efficiency. Therefore, the proposed hybrid scheme is able to obtain better PAPR reduction performance, lower BER, and higher energy efficiency than that of the previous works. Nevertheless, compared with the original OFDM-IM signals, the proposed hybrid scheme still has slight error performance degradation, which is an open topic for future research. 

## Figures and Tables

**Figure 1 entropy-24-01335-f001:**
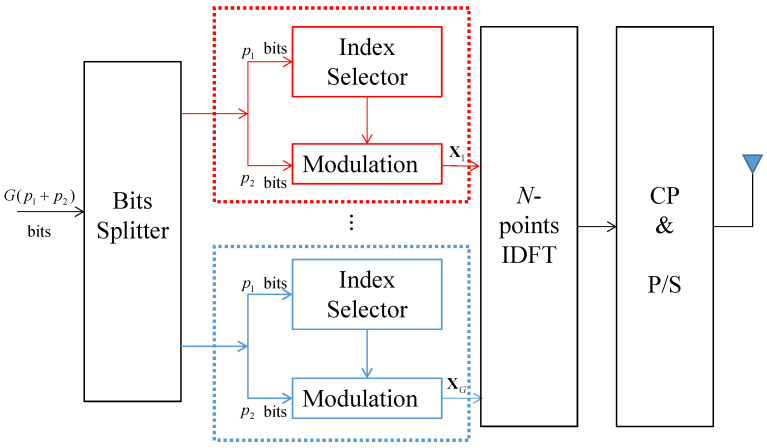
Sub-block grouping of an OFDM-IM system.

**Figure 2 entropy-24-01335-f002:**
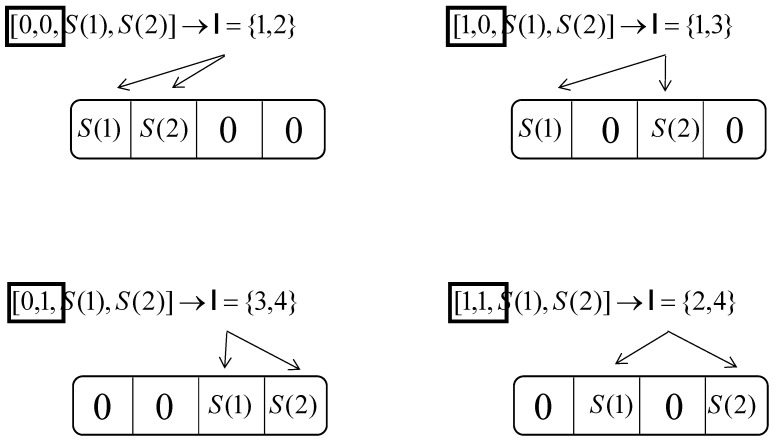
Mapping rules of OFDM-IM when n=4, k=2.

**Figure 3 entropy-24-01335-f003:**
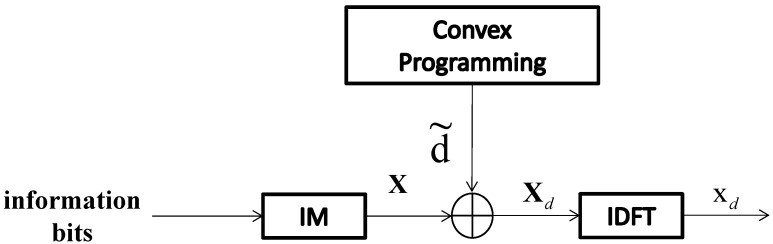
Transmitter structure of the scheme in [17].

**Figure 4 entropy-24-01335-f004:**
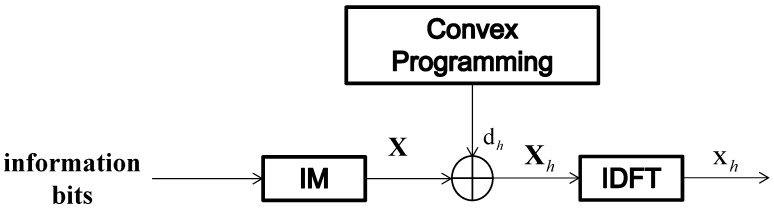
Transmitter structure of the S.1 scheme.

**Figure 5 entropy-24-01335-f005:**
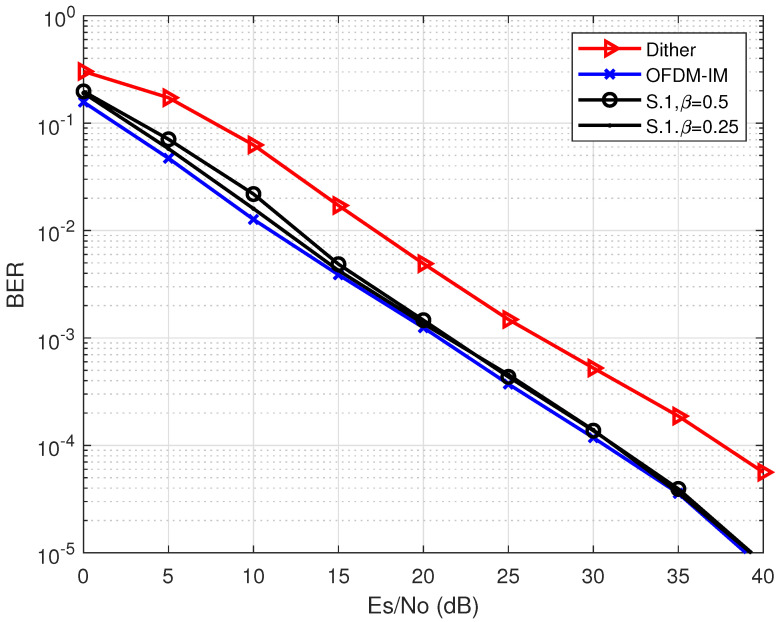
A BER performance comparison among the original OFDM-IM, S.1 scheme, and the Dither scheme [17].

**Figure 6 entropy-24-01335-f006:**
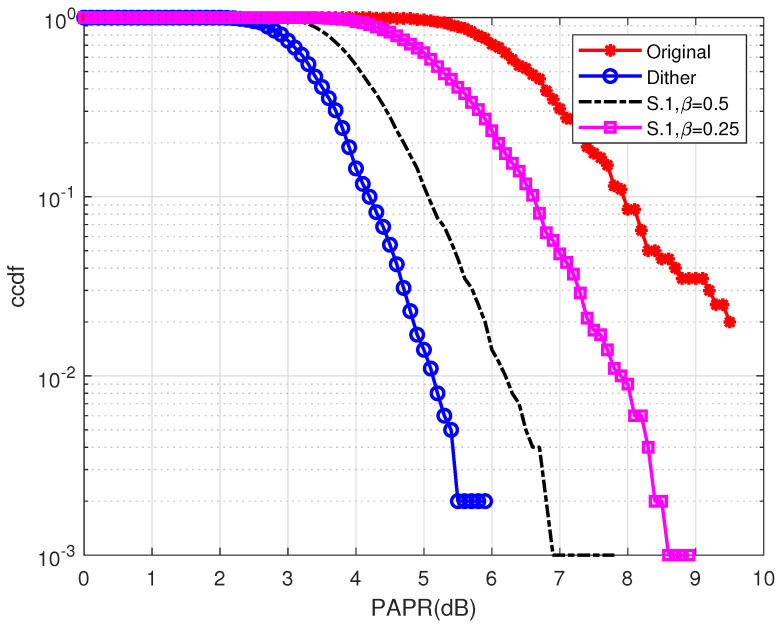
A CCDF comparison among the original OFDM-IM, S.1 scheme, and the Dither scheme [17].

**Figure 7 entropy-24-01335-f007:**
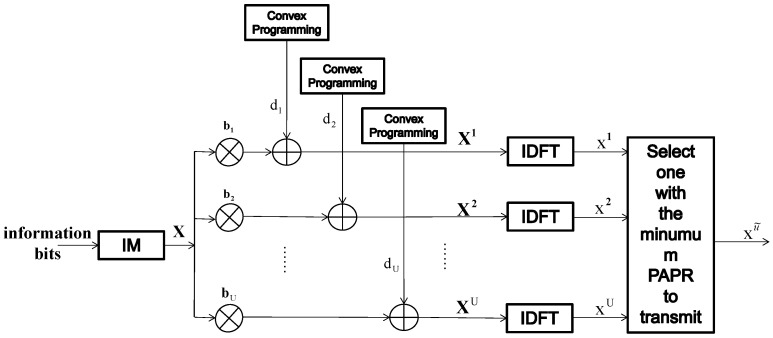
Transmitter structure of the proposed hybrid scheme.

**Figure 8 entropy-24-01335-f008:**
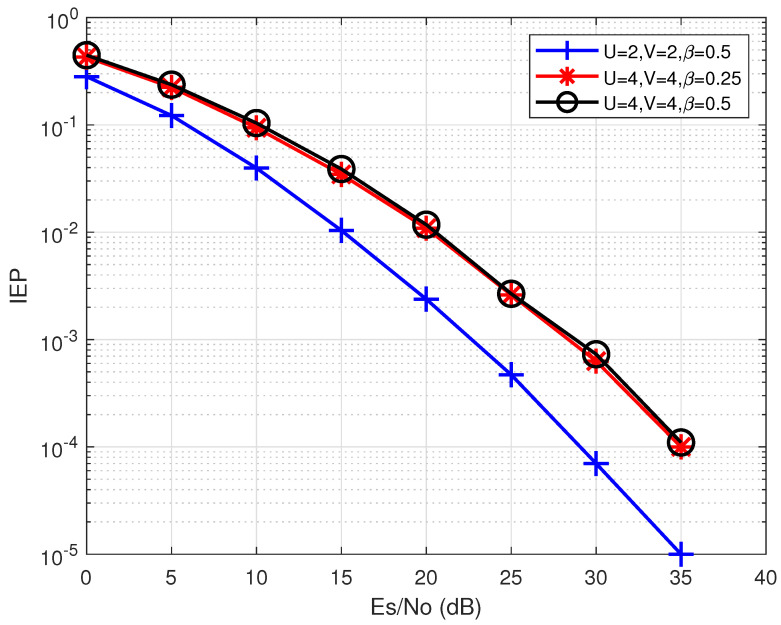
The IER of the proposed hybrid PAPR reduction scheme using different *U*, *V*, and β.

**Figure 9 entropy-24-01335-f009:**
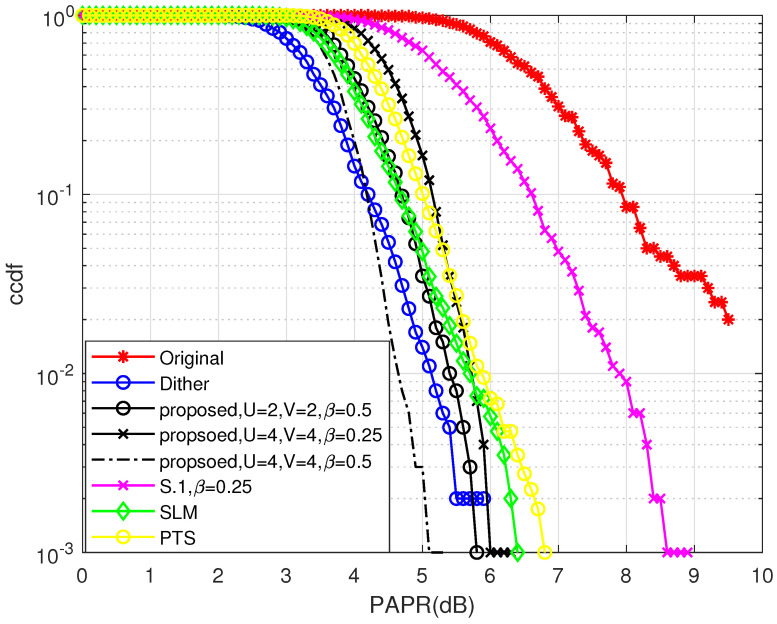
The CCDF performance comparison among the proposed scheme, the SLM scheme, the PTS scheme, the Dither scheme [17], the S.1 scheme, and the original OFDM-IM signal.

**Figure 10 entropy-24-01335-f010:**
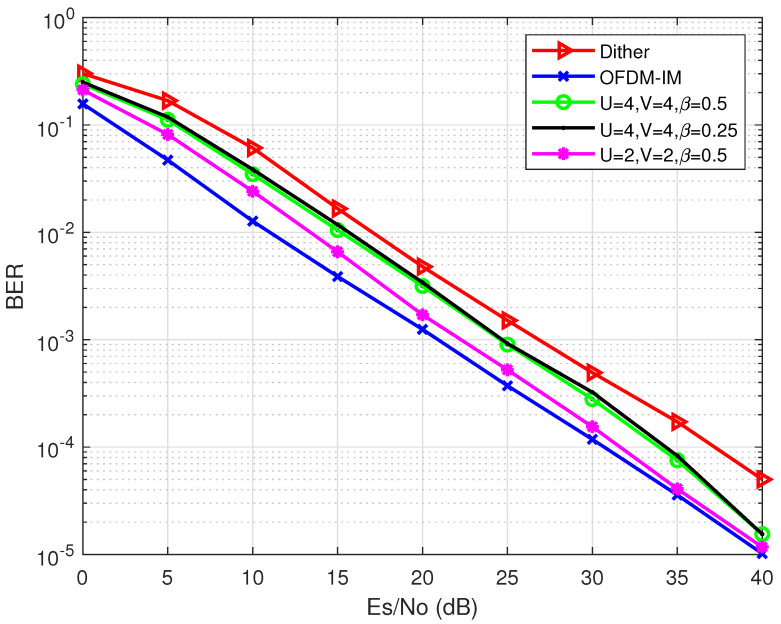
The BER performance comparison among the proposed schemes, the Dither scheme [17], and the original OFDM-IM signal.

**Table 1 entropy-24-01335-t001:** A look-up table example between $b_{u}$ and $\chi_{u}$ for V=4, U=4, and β=0.5.

$b_{1}$	$\chi_{1}$	$\left\{I_{1}^{c},I_{3}^{c}\right\}$
$b_{2}$	$\chi_{2}$	$\left\{I_{2}^{c},I_{4}^{c}\right\}$
$b_{3}$	$\chi_{3}$	$\left\{I_{1}^{c},I_{4}^{c}\right\}$
$b_{4}$	$\chi_{4}$	$\left\{I_{2}^{c},I_{3}^{c}\right\}$

**Table 2 entropy-24-01335-t002:** A candidate construction illustration for V=3, U=2, and β=0.25.

$b_{1}=\left[1,1,1,1\right]$	$\left\{I_{1}^{c},I_{3}^{c}\right\}=\left\{2,4\right\}$	$X^{1}=\left[X\left(0\right),d_{1}\left(0\right),0,d_{1}\left(1\right)\right]$
$b_{2}=\left[-1,-1,-1,-1\right]$	$\left\{I_{2}^{c},I_{3}^{c}\right\}=\left\{3,4\right\}$	$X^{2}=\left[-X\left(0\right),0,d_{2}\left(1\right),d_{2}\left(1\right)\right]$

## Data Availability

Not applicable.

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
