# Peer review of "A Hybrid PAPR Reduction Scheme in OFDM-IM Using Phase Rotation Factors and Dither Signals on Partial Sub-Carriers"

_entropy, 2022, doi:10.3390/e24101335_

Round 1
Reviewer 1 Report
The paper proposed a modified version of OFDM-IM which achieves a tradeoff in performance between peak to average power ratio and bit error rate. The modification is based on an existing method of dither signals by adding phase rotation factors.
The merits of the paper include, 1) The proposed method is sound and the results validate the authors' claim in performance improvement; 2) The paper is well organized and written; 3) The results analysis is thorough.
The weaknesses of the paper are as follows. 1) The paper has minimal improvement over the original method of injecting dither signals. As shown by Fig. 9 and Fig. 10, the proposed method can achieve similar and a little worse PAPR reduction as "Dither" with a little improvement of bit error rate. 2) It's unclear if the detection process described from line 225 to 230 can achieved the optimization described in Eq. (19) or not. If not, how much performance degradation is expected. 3) The complexity of the detection scheme has not been discussed. Can it be done in real time?
Author Response
Dear Reviewer,
Thank you for giving our paper “A Hybrid PAPR Reduction Scheme in OFDM-IM using Phase Rotation Factors and Dither Signals on Partial Sub-carriers” (ID:entropy-1915679) a chance for revision. We have carefully considered all the comments of the editor and reviewers in the new version of the paper as follows:
The 1st comment of Reviewer #1:
The paper has minimal improvement over the original method of injecting dither signals. As shown by Fig. 9 and Fig. 10, the proposed method can achieve similar and a little worse PAPR reduction as "Dither" with a little improvement of bit error rate.
Authors’ Response to the 1st comment of Reviewer #1:
The S.1 scheme in 3.2 can achieve the best BER performance, which is reflected in Fig.5. However, the S.1 scheme has PAPR reduction performance degradation and the S.1 scheme is not practical since it is hard to know instantaneous channel condition.
Therefore, phase rotation factors are introduced thus implements a hybrid PAPR reduction scheme. The hybrid scheme can achieve the best PAPR reduction performance (which is indicated by the black dot line in Fig.9). Moreover, although the hybrid scheme has BER degradation compared with S.1 scheme, its still much better than that of the conventional dither signal based scheme (Fig.10), which is around 5dB when U=V=2 and beta=0.5.
The 2nd comment of Reviewer #1:
It's unclear if the detection process described from line 225 to 230 can achieved the optimization described in Eq. (19) or not. If not, how much performance degradation is expected.
Authors’ Response to the 2nd comment of Reviewer #1:
The detection method described from line 225 to 230 is an energy based detection method proposed for phase rotation factor index detection while the Eq.19 is the optimization process which is used to find the dither signals vector for PAPR reduction. The detection accuracy is shown in Fig.8 and the optimization process for find dither signals is illustrated in [17].
The 3rd comment of Reviewer #1:
The complexity of the detection scheme has not been discussed. Can it be done in real time?
Authors’ Response to the 3rd comment of Reviewer #1:
The detection complexity has been given in the new version of the paper and the corresponding part has been highlighted.
Reviewer 2 Report
A Hybrid PAPR Reduction Scheme in OFDM-IM using Phase Rotation Factors and Dither Signals on Partial Sub-carriers:The topic is current, well-written and manuscript has good grammar. The manuscript is about the peak to average power ratio (PAPR) problem of the OFDM by using Dither signals with phase delay. The proposed method is based on the idea of using the Dither signals with different phases.
The authors should consider the following suggestions provided by the reviewer to improve the scientific depth of their manuscript. They should also address the following comments to improve the quality of the presentation of their manuscript.
1- Abstract is explanatory.
2- In the abstract, Adding the numerical results achieved by the proposed method should be better than existing methods.
3- It is very hard to recognise the contribution of the paper. It should be better to represent the contributions of the paper with a separate paragraph or a subsection.
4- The literature must be strongly updated with some relevant papers focused on the requirements of the 5G within the manuscript [1].
[1] "Waveform design considerations for 5G wireless networks." Towards 5G Wireless Networks-A Physical Layer Perspective (2016): 27-48.
Author Response
Dear Reviewer,
Thank you for giving our paper “A Hybrid PAPR Reduction Scheme in OFDM-IM using Phase Rotation Factors and Dither Signals on Partial Sub-carriers” (ID:entropy-1915679) a chance for revision. We have carefully considered all the comments of the editor and reviewers in the new version of the paper as follows:
The 1st comment of Reviewer #2:
Abstract is explanatory. In the abstract, Adding the numerical results achieved by the proposed method should be better than existing methods.
Authors’ Response to the 1st comment of Reviewer #2:
The abstract has been extended and some numerical results have been cited, which further indicate the superiority of the proposed scheme. The corresponding part has been highlighted in the new version of the paper.
The 2nd comment of Reviewer #2:
It is very hard to recognise the contribution of the paper. It should be better to represent the contributions of the paper with a separate paragraph or a subsection.
Authors’ Response to the 2nd comment of Reviewer #2:
A separate paragraph has been added in order to indicate the contributions of this paper. The corresponding part has been highlighted in the new version of the paper.
The 3rd comment of Reviewer #2:
The literature must be strongly updated with some relevant papers focused on the requirements of the 5G within the manuscript [1]
[1] "Waveform design considerations for 5G wireless networks." Towards 5G Wireless Networks-A Physical Layer Perspective (2016): 27-48.
Authors’ Response to the 3rd comment of Reviewer #2:
A new paragraph has been added focusing on the requirements of the 5G and the corresponding papers have been cited in the new version of the paper. The corresponding part has been highlighted in the new version of the paper.
At the end, we’d like to thank you for the comments and suggestions for improving the quality of the paper and thank you for your consideration of this manuscript.
Sincerely yours,
Si-yu Zhang
